# Equine Colostrum-Derived Mesenchymal Stromal Cells: A Potential Resource for Veterinary Regenerative Medicine

**DOI:** 10.3390/vetsci12070681

**Published:** 2025-07-19

**Authors:** Angelita Capone, Barbara Merlo, Fabiana Begni, Eleonora Iacono

**Affiliations:** 1Department of Veterinary Medical Sciences (DIMEVET), University of Bologna, Ozzano dell’Emilia, 40064 Bologna, Italy; angelita.capone2@unibo.it (A.C.); barbara.merlo@unibo.it (B.M.); fabiana.begni@studio.unibo.it (F.B.); 2IRET Foundation, Ozzano dell’Emilia, 40064 Bologna, Italy; 3Interdepartmental Centre for Industrial Research in Health Sciences and Technology ICIR-HST, University of Bologna, 40064 Bologna, Italy

**Keywords:** equine, colostrum, mesenchymal stem/stromal cells, in vitro differentiation, molecular characterization, veterinary medicine, regenerative medicine

## Abstract

Colostrum, the first nutrient-rich secretion produced by the mammary gland after birth, plays a crucial role in neonatal immune defense and has been shown to contain biologically valuable components, including stem cells. This study investigates the potential of equine colostrum as a non-invasive source of mesenchymal stem/stromal cells (MSCs), which possess regenerative properties. We successfully isolated and expanded these cells in vitro, confirming their mesenchymal identity through key characteristics such as plastic adherence, colony formation, cell–cell adhesion, and the expression of MSC-specific surface markers. Moreover, the isolated MSCs exhibited multilineage differentiation capacity and migration potential, underscoring their suitability for applications in veterinary regenerative medicine. However, considerable variability was observed among the samples, emphasizing the need for validation with a larger sample size. Expanding this research will be essential to assess the reproducibility and therapeutic potential of colostrum-derived MSCs in veterinary medicine, with possible translational implications for human regenerative therapies.

## 1. Introduction

Colostrum is the first, thick, nutrient-rich secretion produced by the female mammary gland during the final 1–2 weeks of pregnancy, as a response to hormonal changes [1,2]. In equine species, due to the epitheliochorial nature of the placenta, colostrum is vital for foal survival, serving as the newborn’s sole natural source of immunoglobulins, fulfilling essential nutritional needs and supporting gastrointestinal tract maturation [2,3,4].

The mammary gland is a highly metabolically active organ that undergoes cyclic phases of proliferation and hypertrophy during pregnancy and lactation, followed by post-lactation involution [5,6,7]. These processes are thought to be supported by mammary stem/stromal cells (MaSCs), which possess the potential to differentiate into the various cellular components essential for lactation [8,9].

The functional unit of the mammary gland is the epithelium, composed of luminal and myoepithelial cells [10,11], both of which are thought to originate from a common progenitor derived from MaSCs [8,9,12]. These MaSCs are likely of epithelial origin and probably present in milk, having been shed from the ductal and luminal epithelial layers. This shedding may result from the increased turnover of secretory tissue or mechanical shear forces induced by the continuous cycles of milk filling and emptying during lactation [7,13]. In 2007, Cregan et al. [14] provided the first evidence that breast milk contains cells with stem cell/progenitor properties. They showed that cell colonies established in culture from breast milk contained cells positive for the mammary stem cell marker CK5 and the general stem cell marker Nestin [14]. The ex vivo presence of CK5- and Nestin-positive cells in breast milk was subsequently confirmed by other investigators [7,15] and further expanded with the identification of breast milk stem cells (BSCs) [16]. BSCs not only exhibit self-renewal capabilities in culture but also express pluripotency-associated genes, including the core transcription factors OCT4, SOX2, and NANOG, along with downstream targets KLF4, REX1, and GDF3 [16]. Remarkably, these BSC subpopulations appear to be highly plastic, capable of differentiating into cell types derived from all three germ layers, including neurons, glia, hepatocytes, pancreatic cells, cardiomyocytes, osteoblasts, chondrocytes, and adipocytes [10,16,17,18].

In addition to BSCs, various types of stem cells, including MSC (mesenchymal stem/stromal cell)-like cells, have been identified in colostrum and milk across different species [7,10,12,16,19,20,21,22,23,24,25,26]. Patki et al. [10] confirmed the mesenchymal characteristics of human breast milk-derived stem cells, which exhibit dual properties by being positive for both the epithelial marker E-Cadherin and the mesenchymal marker SMA (*Smooth Muscle Actin*), supporting the hypothesis that these cells may be of myoepithelial origin [10]. Moreover, these human breast milk-derived MSCs have been shown to possess the ability to differentiate into various human cell types, underscoring their potential applications in regenerative medicine [10].

These findings pave the way for further research into the use of milk and colostrum-derived stem cells as a non-invasive, accessible source for therapeutic applications in regenerative medicine, including in veterinary medicine. In vivo, these stem cells contribute not only to the structural remodeling of the mammary gland and the development of neonatal tissues but also play a role in epigenetic regulation in offspring [26]. Studies in rats and mice demonstrated that maternal stem cells can survive in the gastrointestinal tract, enter the bloodstream, and integrate into various neonatal organs, where they differentiate into functional cells in a process known as microchimerism [27,28]. This process, which begins in utero and continues throughout breastfeeding, suggests that milk- and colostrum-derived stem cells not only support neonatal development but may provide long-term immune benefits and foster unique biological tolerance between mother and offspring [12,29].

Among stem cell types, MSCs are widely used for cell therapy due to their advantages, including self-renewal, multi-lineage differentiation, the low risk of teratoma formation, and low immunogenicity [30,31]. MSCs can be derived from various tissues, such as bone marrow [32], adipose tissue [33], Wharton’s jelly [34], placenta [35], the amniotic membrane [36], and amniotic fluid [37]. In 2006, the International Society for Cell & Gene Therapy (ISCT) established the “golden criteria” for MSC characterization. These criteria include a fibroblast-like plastic-adherent morphology, a high expression level of mesenchymal biomarkers (CD73, CD90, and CD105), minimal expression of hematopoietic or immune biomarkers (e.g., CD31, CD34, CD45, and HLA-DR), and the ability to differentiate into adipocytes, osteoblasts, and chondrocytes [38]. Moreover, in response to stimuli, MSCs secrete a plethora of bioactive molecules with anti-inflammatory [39,40], immunomodulatory [39,41,42,43], and other therapeutic effects, such as inhibiting cell death, reducing fibrosis, promoting vascularization, and exhibiting antimicrobial properties [42,44,45].

Over the past two decades, studies have extensively characterized human milk-derived stem cells [5,6,7,10,14,16,18,22,23,25,46,47,48,49,50,51,52] and their therapeutic potential across various pathological conditions [53,54,55,56,57,58].

In veterinary species, studies have primarily focused on isolating and characterizing MSC-like cells from bovine, swine, and rabbit milk [19,59,60], highlighting their regenerative, pro-angiogenic [61], and antimicrobial [60] properties with potential applications in neonatal health and veterinary medicine.

However, research on equine mammary and milk-derived stem cells remains limited. To date, only one study has successfully isolated and characterized mammary stem cells (MaSCs) from equine mammary gland tissue [62]. Subsequent investigations have explored the role of extracellular vesicles [63] and microRNAs [64] in MaSC regulation, the development of equine mammary organoids for comparative research [65], and species-specific mechanisms of mammary cancer resistance [66].

Given the established bioactivity and immunological relevance of colostrum, we hypothesize that equine colostrum may contain mesenchymal stem/stromal cells (MSCs) with regenerative potential. This study aims to isolate and characterize MSCs for the first time from equine colostrum collected immediately after delivery, laying the groundwork for future applications in veterinary regenerative medicine.

## 2. Materials and Methods

### 2.1. Materials

Chemicals were obtained from Sigma-Aldrich (Merck KGaA, Darmstadt, Germany). FBS (fetal bovine serum) was purchased from GIBCO™ (Thermo Fisher Scientific, Waltham, MA, USA). Plastics were purchased from Falcon™ (Corning Inc., New York, NY, USA) and SarstedtTM (Sarstedt AG & Co. KG, Nümbrecht, Germany), unless otherwise stated.

### 2.2. Samples

Colostrum samples were collected from 6 healthy standardbreed mares hospitalized at the Equine Perinatology and Reproduction Unit of the Equine Clinical Service, Department of Veterinary Medical Sciences, University of Bologna (Italy), for delivery management. All the mares had a normal pregnancy concluded with an uncomplicated delivery and the birth of a foal with an Apgar score of 10 at birth [67]. Mares were housed in wide straw bedding boxes and fed with hay *ad libitum* and concentrates twice a day. During the day, mares were allowed to go to pasture. Detailed data on the age and number of pregnancies of the donor mares are summarized in Table 3.

Samples were aseptically collected by manual stripping in sterile 15 mL tubes immediately after foaling. Colostrum quality, indicative of its IgG concentration [68], was assessed using a refractometer equipped with a Brix scale [68,69]. The portion of each sample not utilized for quality assessment was allocated for cell isolation. It was stored at +4 °C for a maximum of 12 h before being transported to the Laboratory of Animal Reproduction and Biotechnologies (Equine Clinical Service, Department of Veterinary Medical Sciences, University of Bologna, Italy). All procedures were conducted in compliance with the guidelines approved by the Animal Welfare Committee (CoBA) of the University of Bologna (protocol no. 4630/23).

### 2.3. Equine Colostrum-Derived MSCs’ Isolation and Culture

Equine colostrum-derived MSCs (C-MSCs) were isolated as previously described by Widjaja et al. [70] with some modifications. Briefly, colostrum was mixed, diluted at a ratio of 1:1 with Dulbecco’s polyphosphate-buffered saline (DPBS) plus antibiotics (100 IU/mL penicillin, 100 μg/mL streptomycin) and antimycotics (2.5 mg/L Amphotericin B) in 50 mL conical tube, and centrifuged at 1881× *g* for 20 min at 20 °C. After centrifugation, the supernatant and the fat layer were discarded, and the cell pellet was re-suspended in 2 mL DPBS, then transferred into a sterile 15 mL conical tube. The cell pellet was re-suspended and centrifuged at 1881× *g* for 10 min at 20 °C. This operation was repeated twice. Finally, the cellular pellet was re-suspended in complete culture medium (DMEM F-12 + 10% FBS + 100 IU/mL penicillin + 100 μg/mL streptomycin). Cells were plated into 25 cm^2^ culture flasks and incubated in 5% CO_2_ at 38.5 °C in a humidified atmosphere (Passage 0). After 24 h, the culture medium was completely replaced, and non-adherent cells were removed. The culture medium was changed every 3 days until reaching a confluence of 80–90%. At this time, cells were dissociated using a 0.25% Trypsin-EDTA (ethylenediaminetetraacetic acid) solution, counted using a Neubauer improved chamber and cryopreserved as described by Merlo et al. [71]. Briefly, cells in 0.5 mL of FBS were put in a 1.5 mL cryogenic tube at 4 °C. After 10 min, the cell suspension was diluted at a ratio of 1:1 with FBS + 16% DMSO (dimethyl sulfoxide, with a final concentration of 8%) and maintained for a further 10 min at 4 °C. Then, the cryogenic tube was set to −80 °C for 24 h in Mr. Frosty (Nalgene, Rochester, NY, USA) and finally stored in liquid nitrogen.

Equine C-MSCs were thawed at 37 °C in 20 mL DMEM + 10% FBS, then centrifuged at 470× *g* at 25 °C for 10 min. The pellet was re-suspended in 1 mL of complete culture medium, and cell concentration and viability were evaluated by staining cells with Trypan blue (0.4%) solution. Cells were seeded in a 25 cm^2^ flask (5000 cells/cm^2^) as “Passage 1” (P1). Passage 3 (P3) cells were used for in vitro tests.

### 2.4. Characterization of Colostrum-Derived MSCs

When a confluence of 80–90% at P2 was reached, cells from all six colostrum samples were detached from the flask, and viability and concentrations were determined as described above. The cells were subsequently seeded to evaluate their growth rate through population doubling time (PDT) analysis and for characterization via a colony-forming unit (CFU) assay, adhesion and migration assay, and tri-lineage in vitro differentiation. Furthermore, CD marker expression was evaluated by RT-PCR. For all samples, each test was carried out with three replicates.

#### 2.4.1. Population Doubling Time (PDT) Analysis

To determine growth rate, C-MSCs were seeded in 6-well plate at a cell density of 5000 cells/cm^2^. Cell number was assessed every day for 7 days, and the assessments were performed in triplicate. The mean number of cells was calculated and plotted on a semi-log curve against culture time to generate a growth curve. The population doubling time (PDT) was calculated from the log phase of the growth curve using the equation described by Bezerra et al. (2022) [72]:PDT = (t × log2)/log (N*t* / N0)(1)
where t represents the period of cultivation in days, N0 represents the initial number of cells, and N*t* represents the number of cells in a specific period of culture.

#### 2.4.2. CFU (Colony-Forming Unit) Assay

For determining the ability of cells to form colonies, 1 × 10^2^ cells were cultured for 4 weeks in 6-well plates. Colonies were then fixed in 4% paraformaldehyde at room temperature (RT) for 1 h and stained with Giemsa 0.1% stain (30 min). Using an inverted light microscope (Eclipse TE 2000-U, Nikon Instruments Inc., Tokyo, Japan), the operator counted the number of colonies formed by at least 16–20 nucleate cells in triplicate. Colonies were categorized as “dispersed,” characterized by a vague macroscopic appearance and scattered cells under the microscope, as opposed to dense, compact colonies with the typical “fingerprint” pattern [73].

#### 2.4.3. Adhesion and Migration Assays

To determine whether C-MSCs preserved their adhesion capability, a spheroid formation assay was performed. Differently from cell–substratum adhesion assay, which is performed on monolayer cultures adherent to rigid substrates, this test gives information about the direct cell–cell adhesion architecture found in normal tissues. Cells were cultured through the ‘hanging-drop’ method in Corning spheroid microplates (96-well, black/clear bottom round, ultra-low attachment surface; Corning Inc., NY, USA; 5000 cells/25 µL drop). Bright-field images were acquired by a CCD camera (DS-Fi2, Nikon, Tokyo, Japan) mounted on an inverted light microscope (Eclipse TE 2000-U, Nikon Instruments Inc., Tokyo, Japan) using 4× magnification every 24 h from seeding to the formation of a complete spheroid after 96 h. Starting from the binary masks obtained by Fiji software (ImageJ, v. 1.8.0_345/1.54 g), the volume of each spheroid was computed using *ReViSP* (sourceforge.net/projects/revisp/, v. 2.3) [74], a software specifically designed to accurately estimate the volume of spheroids and to render an image of their 3D surface.

Given the growing interest in the role of MSCs in tissue repair [75], a scratch assay (or wound-healing assay) was carried out to assess their migration potential, as previously described by Liang et al. [76]. Briefly, P3 cells were seeded (5000 cells/cm^2^) in 6-well plates. At 80 to 90% confluence, the cell monolayer was scraped using a 200 μL pipette tip. The cells were then washed twice with DPBS to remove any detached cells, and the plate was incubated under standard culture conditions. Images were captured at the same two locations of each well, both immediately after scraping the cells (time 0 = t*0*) and every 24 h over a 96 h period (0, 24, 48, 72, 96 h). The distances of each scratch closure were measured by Fiji software (ImageJ, v. 1.8.0_345/1.54 g), and the migration percentages (*Wound Closure* %) were calculated using the following formula [77]:[(*distance at t*0 − *distance at tn* ∗ 100]/*distance at t*0(2)
where t*n* corresponds to the distance at each specific time point analyzed.

#### 2.4.4. Multi-Lineage In Vitro Differentiation

C-MSCs’ in vitro differentiation potential toward osteogenic, adipogenic, and chondrogenic lineages was also evaluated. Cells were seeded in 24-well plates at a density of 5000 cells/cm^2^ and cultured until subconfluency was achieved. Then, the culture medium was replaced, and cells were cultured for 3 weeks under specific induction media (Table 1) [78]. As a negative control, an equal number of cells were cultured in expansion medium. The specific induction media and expansion medium were replaced twice a week. To assess differentiation, cells were fixed with 4% paraformaldehyde at RT for 1 h. Differentiation was visualized using Oil Red O for adipogenic vacuoles, Alcian Blue for sulfated proteoglycans, and Alizarin Red S for calcium deposits. Bright-field images were captured using a CCD camera (DS-Fi2, Nikon, Tokyo, Japan) mounted on an inverted light microscope (Eclipse TE 2000-U, Nikon Instruments Inc., Tokyo, Japan) at 10× magnification.

Quantification of positively stained areas was performed using Fiji software (ImageJ, v. 1.8.0_345/1.54 g) following the methodology described by Heyman et al. (2022) [79]. The “*ColourDeconvolution2*” plugin was applied to separate the stain and background channels, which were then converted to 8-bit grayscale images and binarized using auto-thresholding. The positively stained areas were measured, and the differentiation ratio (DR) was calculated as the ratio between the areas of the stain-positive signal and the background.

To compare differentiation across samples, a normalized DR was determined for each sample by dividing the mean DR of the differentiated wells by the mean DR of the non-induced controls.

#### 2.4.5. Molecular Characterization

For the molecular characterization of C-MSCs, total RNA was extracted from snap-frozen cells using the NucleoSpin® RNA Kit (Macherey-Nagel, Düren, Germany) following the manufacturer’s protocol. Residual genomic DNA was removed by treating the RNA with Amplification-Grade DNase I (Invitrogen, Thermo Fisher Scientific, Waltham, MA, USA). Complementary DNA (cDNA) was synthesized from purified RNA using the RevertAid Reverse Transcriptase Kit (Thermo Fisher Scientific, Waltham, MA, USA).

The resulting cDNA was directly subjected to PCR using the 2× EpiArt™ HS Taq Master Mix (Vazyme Biotech Co., Ltd., Nanjing, China). The expression of genes encoding mesenchymal stem/stromal cell markers (CD90 and CD73), hematopoietic markers (CD45 and CD34), and major histocompatibility complex markers (MHC-I and MHC-II) was analyzed. GAPDH was used as a reference gene, as its stable expression has been previously validated in our laboratory under similar experimental conditions [80,81,82]. PCR amplification was carried out using 34 cycles with the following thermal profile: initial denaturation at 95 °C for 5 min; 34 cycles of denaturation at 95 °C for 30 s, annealing at 58 °C for 30 s, and extension at 72 °C for 45 s; and a final extension at 72 °C for 5 min. The specific primers employed for amplification are listed in Table 2 [83,84,85].

PCR products were resolved on a 2% agarose gel stained with ethidium bromide. Electrophoresis was conducted to verify the expected band sizes, and gels were visualized and photographed under ultraviolet light using a UVP PhotoDoc-It Imaging System (UVP, LLC, Upland, CA, USA). The presence of discrete bands corresponding to the expected sizes confirmed positive expression, while the absence of bands indicated a negative result. A 50 bp DNA Ladder (Thermo Fisher Scientific, Waltham, MA, USA) was used as a molecular size marker.

#### 2.4.6. Statistical Analysis

Data are reported as mean ± standard deviation (SD). Statistical analyses and graph generation were performed using GraphPad Prism software (version 9.0.0). Data were analyzed for normal distribution using the Shapiro–Wilk test. Population doubling times (PDTs) and colony numbers were analyzed using one-way ANOVA, followed by Tukey’s post-hoc test for pairwise comparisons. Proliferation rates and wound closure percentages were evaluated with two-way ANOVA, followed by Tukey’s post-hoc test for comparisons between groups. Volumes of 3D spheroids at 96 h and differentiation ratio (DRs) were analyzed using the Kruskal–Wallis test, followed by Dunn’s multiple comparisons test. To evaluate a potential correlation between maternal factors (age and parity) and cell behavior (cell yield and PDT), Pearson’s correlation test was performed. Results were considered significant when the probability of their occurrence due to chance alone was less than 5% (*p* < 0.05).

## 3. Results

### 3.1. Isolation and Morphological Characterization of Equine Colostrum-Derived Cells

Equine colostrum cells (*n* = 6; ~20 mL/sample) were successfully isolated following the protocol detailed in the Section 2. Their isolation was based solely on the cells’ ability to adhere to plastic. Phase-contrast microscopy (Figure 1; The original image is published in the Appendix A) revealed a morphologically heterogeneous population, comprising cells with a round/polyhedral, epithelial-like appearance, alongside spindle-shaped cells characteristic of MSCs. Cell colonies became visible within the first 24 h and reached confluence in 10.17 ± 1.17 days, with a cell yield ranging from 30 to 500 × 10^3^ cells/mL of colostrum.

A detailed summary of the colostrum samples, including volume, Brix index score at the time of collection, number of cells isolated, and time required to reach confluence, is reported in Table 3. A statistically significant negative correlation was found between the number of previous parturitions (parity) and the number of cells isolated per mL of colostrum (*r* = −0.833; *p* < 0.05). In contrast, no significant correlation was observed between the age of the mare and the cell yield (*p* > 0.05).

Undifferentiated cells were cultured up to Passage 3 (P3) across all samples, with no observable changes in morphology during the culture period or after cryopreservation and thawing. The post-thaw viability, determined by Trypan Blue staining, at P3 exceeded 70%, while the initial viability was over 95%.

### 3.2. Characterization of Colostrum-Derived MSCs

#### 3.2.1. Population Doubling Time (PDT) Analysis

To assess the growth efficiency and proliferation rate of successfully established colostrum-derived MSC (C-MSC) cultures, population doubling times (PDTs) were calculated. The semi-logarithmic growth curves (Figure 2; The original image is published in the Appendix A) revealed variability in the growth patterns across the samples. Three samples (4–5-6) exhibited a brief lag phase of 1–2 days, followed by a significant and linear increase in cell concentration, while three samples demonstrated prolonged lag phases (1–2) or a slower overall growth rate (3). The mean PDT values, which confirm the variability between samples, are summarized in Table 3 and illustrated in Figure 2. One sample (3) exhibited the longest PDT (6.21 ± 1.84 days), indicating reduced proliferative efficiency, which was significantly different from the mean PDT of the other sample (2.0 ± 0.7 days, *p* < 0.05; Figure 2). In contrast, two out of six samples (2–4) showed the shortest PDTs, reflecting a higher proliferation rate (1.42 ± 0.28 and 1.22 ± 0.08, respectively). Overall, the effect on both the growth rate and PDT was highly significant (*p* < 0.05).

No significant correlation was observed between the population doubling times (PDTs) and either the age of the mares or their parity (*p* > 0.05).

#### 3.2.2. CFU (Colony-Forming Unit) Assay

The colony-forming unit (CFU) assay was performed to evaluate the self-renewal capacity of C-MSCs. The average number of colonies formed is summarized in Table 3 and presented in Figure 3, revealing significant sample-related variability (*p* < 0.05). Two samples (2–4) demonstrated the highest clonogenic potential, forming significantly more CFUs than the other samples (71.0 ± 16.7 and 59.3 ± 7.37, respectively; *p* < 0.05). Conversely, one sample (6) exhibited the lowest CFU count (13 ± 3.6), showing a statistically significant difference compared to samples 2, 4, and 5 (*p* < 0.05, Figure 3; The original image is published in the Appendix A). No significant differences were detected among samples 1, 3, and 5 (21.0 ± 2, 34.7 ± 11.0, and 40.3 ± 5.5, respectively; Figure 3).

#### 3.2.3. Adhesion and Migration Assays

At Passage 3 (P3), using a hanging drop culture system, all tested MSC populations, except for samples 1 and 3, formed three-dimensional spheroid-like structures, demonstrating effective cell–cell adhesion.

Spheroid formation was evident after 24 h, and by the fourth day of incubation, most C-MSCs had developed stable, compact, and rigid spheroids with a spherical morphology (Figure 4; The original image is published in the Appendix A).

In contrast, samples 1 and 3 failed to generate spheroids, instead forming loosely compact cellular sheets that remained easily dissociable, suggesting a lack of spheroid-forming capacity (Figure 4). Bright-field images were acquired at multiple time points (24, 48, 72, and 96 h) and analyzed using *ReViSP* (sourceforge.net/projects/revisp/, v. 2.3), an open-source voxel-based 3D image analysis tool capable of estimating spheroid volume starting from binary masks and rendering a three-dimensional surface representation (Figure 4; The original image is published in the Appendix A).

A quantitative analysis of spheroid volume revealed dynamic structural changes over time, characterized by progressive compaction and a corresponding reduction in volume between 48 and 96 h. By the end of the incubation period (4 days), spheroid volumes had stabilized, with no significant differences detected between the samples.

To evaluate the migratory potential of C-MSCs, we measured the wound closure percentage at 24, 48, 72, and 96 h (Figure 5; The original image is published in the Appendix A).

The quantitative analysis revealed significant differences in wound closure percentages among the groups (two-way ANOVA, *p* < 0.05). Notably, sample 2 demonstrated the highest migratory capacity, achieving complete wound closure (100%) at 72 h, significantly outperforming all other groups. In contrast, samples 1, 3, 4, and 6 exhibited a more gradual increase in wound closure over time, without exceeding approximately 30% at 96 h (Figure 5).

Inter-group comparisons at individual time points revealed intermediate wound closure dynamics for sample 5. Although its performance showed modest but non-significant improvements over the slower-migrating groups (1, 3, 4, and 6), this remained significantly lower than that of sample 2 (*p* < 0.05, Figure 5).

#### 3.2.4. Multi-Lineage In Vitro Differentiation

C-MSCs were assessed for their trilineage differentiation potential (adipogenic, chondrogenic, and osteogenic), in accordance with ISCT guidelines.

After 3 weeks of adipogenic induction, all isolated cells exhibited adipogenic differentiation, as demonstrated by a morphological transition from spindle-shaped to round or oval-shaped cells. This was accompanied by the presence of intracytoplasmic lipid droplets, confirmed by positive staining with Oil Red O (Figure 6; The original image is published in the Appendix A). In contrast, control MSCs maintained their spindle-shaped morphology, forming a monolayer without staining with Oil Red O, confirming the absence of adipogenic differentiation (Figure 6).

Chondrogenic differentiation was observed in the C-MSCs from all samples after 21 days of culture in a chondrogenic medium. This was marked by a morphological change from spindle-shaped cells to larger, round cell aggregates, along with the secretion of an extracellular matrix enriched in sulfated proteoglycans, a defining trait of cartilage tissue. Positive Alcian Blue staining (Figure 6) confirmed the presence of sulfated proteoglycans, while no staining was detected in the control wells (Figure 6), indicating a lack of chondrogenic differentiation.

Following osteogenic induction, C-MSCs from all samples exhibited matrix calcium deposition, as demonstrated by positive Alizarin Red S staining, a key marker of osteogenic differentiation (Figure 6). No calcium deposition was observed in the control groups (Figure 6).

The quantification of positively stained areas using ImageJ software showed no statistically significant differences in adipogenic, chondrogenic, or osteogenic differentiation capacities between the groups (*p* > 0.05).

#### 3.2.5. Molecular Characterization

A gene expression analysis at P3 was performed to characterize C-MSCs in terms of mesenchymal (CD90 and CD73), hematopoietic (CD45 and CD34), and immunogenic (MHC-I and MHC-II) markers. GAPDH was used as a housekeeping gene (Figure 7; The original image is published in the Appendix A). The qualitative PCR results showed that all C-MSC populations expressed the mesenchymal markers CD90 and CD73, while CD34 and CD45 were not detected. MHC-I expression was weakly positive in all samples except for sample 1, in which it was absent. MHC-II expression was consistently negative across all samples. The results are summarized in Table 4.

## 4. Discussion

The expanding field of regenerative medicine has intensified the search for innovative and advantageous stem cell sources. Recent studies have demonstrated that milk, including colostrum, harbors a heterogeneous population of cells, including stem and progenitor cells, with significant regenerative potential [5,6,7,10,14,16,18,22,23,25,46,47,48,49,50,51,52].

While interest in the therapeutic applications of milk- and colostrum-derived stem cells in veterinary medicine is increasing [19,59,60,61], little is known about the presence and characteristics of mesenchymal stem/stromal cells in equine colostrum.

This study aimed to isolate and characterize equine colostrum-derived MSCs (C-MSCs) to validate colostrum as a promising, non-invasive stem cell source. Building on our previous findings on fetal- and adipose-derived MSCs [80,82], we sought to compare C-MSCs with these well-characterized sources, assessing their potential for veterinary and translational medicine.

Our findings confirmed the successful isolation of C-MSCs, with a cell yield ranging from 30 to 500 × 10^3^ cells/mL. These cells exhibited plastic adherence and a fibroblast-like morphology, consistent with prior reports on MSCs isolated from human breast milk [47,86]. The presence of a heterogeneous cell population in colostrum, including both fibroblast-like and round-shaped cells, aligns with studies on bovine and human milk-derived MSCs, in which an epithelial-to-mesenchymal transition (EMT) has been observed during culture [10,47,59].

Studies on human breast milk-derived MSCs have demonstrated that the stage of lactation significantly affects stem cell yield and viability, with higher concentrations typically observed during early lactation [6,47,48]. Although further research is required in the equine species to directly compare early versus late lactation phases, our findings indicate that MSCs can be reliably isolated from mammary secretions collected immediately postpartum.

Beyond the influence of lactation stage, our data suggest that maternal parity, rather than age, significantly affects the yield of MSCs isolated from equine colostrum. Despite the limited sample size, a statistically significant negative correlation was observed between the number of previous parturitions (parity) and the cell yield. This suggests that a higher number of pregnancies may be associated with a reduced recovery of adherent cells from colostrum.

Comparable observations have been reported in human studies, in which a reduced percentage of CD105^+^ cells in mature milk was associated with increasing parity, possibly due to cumulative histological and anatomical changes in the mammary gland over successive pregnancies [48].

Taken together, our results support the hypothesis that the early postpartum period, particularly in low-parity mares, may offer a more favorable window for MSC recovery. This advantage likely reflects both the biological characteristics of early lactation and the structural remodeling of mammary tissue associated with successive pregnancies.

MSC characterization was performed at Passage 3, when the culture reached a homogeneous population, in accordance with previous studies [87,88,89,90]. The population doubling time (PDT), a key parameter for assessing MSC proliferative potential in therapeutic applications [90,91], exhibited notable inter-sample variability. While some colostrum-derived MSC (C-MSC) samples expanded rapidly, others proliferated more slowly, with an average PDT of 2.7 ± 0.6 days. This variability aligns with existing reports highlighting the influence of tissue origin and culture conditions on MSC proliferation [92,93]. The proliferation rate observed for C-MSCs is comparable to that reported for Wharton’s jelly MSCs (WJ-MSCs, 2.7 ± 0.9 days) [82]. In contrast, C-MSCs proliferated more rapidly than bone marrow MSCs (BM-MSCs, 3.6 ± 1.4 days) [82], but at a slower rate than MSCs derived from amniotic fluid (AF-MSCs, 27.13 ± 2.6 h) [92], adipose tissue MSCs (AT-MSCs, 2.2 ± 1.1 days), and umbilical cord blood MSCs (UCB-MSCs, 2.4 ± 1.3 days) [82].

Although age-related changes in PDT have been documented in other studies [94,95], the relatively small number of donors may have limited our ability to detect such effects, as previously suggested by Vidal et al. [96].

Interestingly, parity did not significantly influence PDTs, indicating that the intrinsic proliferative properties of isolated MSCs remain consistent regardless of the number of previous pregnancies. This observation implies that while parity may impact the initial quantity of recoverable cells, it does not appear to alter their in vitro growth kinetics. Similar findings have been reported in studies in which MSCs from donors with different parity histories showed no significant differences in expansion potential under standardized culture conditions [97].

The colony-forming unit (CFU) assay confirmed the self-renewal capacity of C-MSCs. Our results contrast with those reported by Fan et al.; indeed, the authors failed to obtain CFUs from human breast milk MSCs, possibly due to differences in lactation stage, culture conditions, or intrinsic biological differences between species [7]. The ability of C-MSCs to form colonies suggests a robust self-renewal potential, comparable to that of MSCs derived from equine fetal annexes and other postnatal sources [98]. Further comparative studies are required to elucidate the specific factors influencing CFU efficiency across different species and MSC sources.

Spheroid formation was assessed using the hanging drop method, with most samples forming compact spheroids indicative of strong cell–cell adhesion properties. Notably, smaller spheroids correlated with an increased adhesion ability, a characteristic previously described in fetal-derived MSCs, such as WJ-MSCs, compared to adult tissue-derived MSCs [82]. This suggests that C-MSCs share adhesion properties similar to those observed in fetal MSCs.

The migratory potential of MSCs is a key characteristic that influences their capacity for systemic application and integration into host tissues during therapeutic interventions [30]. The scratch assay demonstrated efficient C-MSC migration, achieving approximately 30% wound closure at 96 h. Comparative studies in our laboratory showed similar migration efficiencies among MSCs from various sources, including the amniotic membrane (AM-MSCs, 34.14 ± 4.51%) and Wharton’s jelly (WJ-MSCs, 38.20 ± 2.88%) [80]. Comparable migration rates were observed among MSCs from WJ, UCB, BM, and AT, though AT-MSCs exhibited notable inter-sample variability [82]. These findings indicate that C-MSCs possess migration capabilities like fetal annex-derived MSCs, reinforcing their regenerative potential.

C-MSCs exhibited a multipotent differentiation capacity, successfully differentiating into osteogenic, chondrogenic, and adipogenic lineages, as confirmed by specific staining performed after 21 days of culture in the appropriate differentiation medium. These results align with prior studies on MSCs from human breast milk [10,16] and bovine milk [59], which demonstrated similar tri-lineage differentiation potential. The mesodermal differentiation ability of C-MSCs further validates their identity as MSCs, in accordance with the criteria established by the International Society for Cell & Gene Therapy (ISCT) [38]. Notably, unlike MSCs from adult equine tissues [99] and our previous findings on equine AF-MSCs, UCB-MSCs, and WJ-MSCs [88], C-MSCs displayed minimal donor-to-donor variability.

To confirm the mesenchymal identity of equine C-MSCs, we evaluated MSC surface marker expression. These cells expressed CD90 and CD73 while lacking hematopoietic markers CD34 and CD45, consistent with findings on MSCs from AF, UCB, and WJ [88]. Similar expression patterns have been reported in MSCs from milk across species, including rabbits, bovines, and humans, with high levels of CD29, CD166, CD44, and CD105 and an absence of CD45 and CD34 [10,19,22,47,58,59,100].

A key feature of MSCs is their immunomodulatory potential, which is largely influenced by the expression of major histocompatibility complex (MHC) molecules. In our study, MHC-I expression varied among samples, with some exhibiting weak expression and one lacking it entirely. Previous studies on equine BM-MSCs have shown MHC-I expression with heterogeneous MHC-II expression [101,102]. In horses, MSCs that express MHC-I but not MHC-II have been shown to avoid triggering T-cell proliferation [102]. The absence of MHC-II expression is therefore a crucial factor for potential allogeneic applications, as it enables MSCs to evade immune rejection [102,103]. Our findings confirm that equine C-MSCs did not express MHC-II, in agreement with our previous studies on equine WJ- and AT-MSCs [80] and recent findings on AF-MSCs [92].

The successful isolation and characterization of equine C-MSCs highlight their potential as a novel, non-invasive source of regenerative cells for veterinary medicine. Their ease of collection, multipotency, and immunomodulatory properties position them as strong candidates for equine regenerative therapies, with possible translational implications for human medicine. However, the variability observed among samples from different donors necessitates validation with a larger sample size. Future studies should also investigate their in vivo therapeutic potential and interactions with host tissues.

## 5. Conclusions

This study provides the first comprehensive characterization of mesenchymal stem/stromal cells isolated from equine colostrum, confirming their mesenchymal phenotype, self-renewal ability, migration potential, and multipotency. These findings pave the way for further investigations into their applications in veterinary regenerative medicine, with potential translational implications for human therapies.

## Figures and Tables

**Figure 1 vetsci-12-00681-f001:**
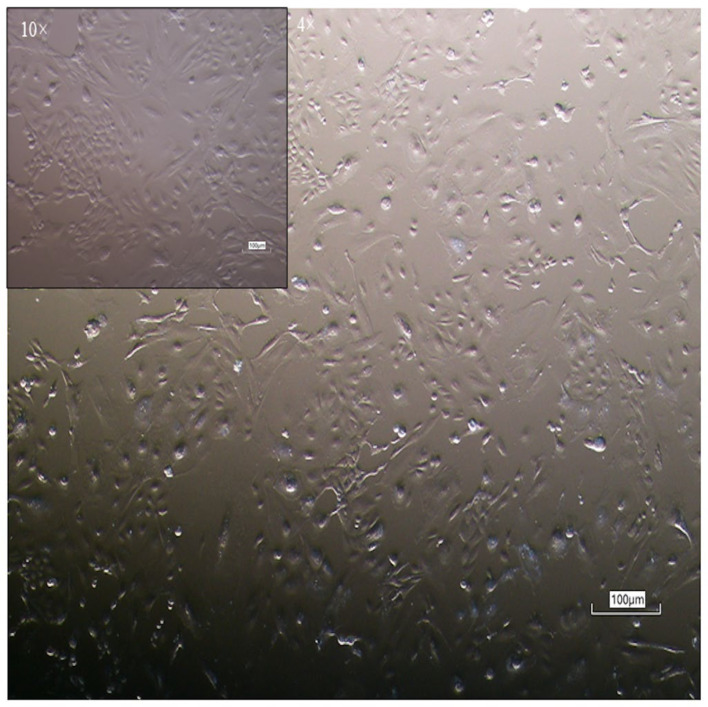
Monolayer of equine colostrum-derived MSCs; magnifications of 40× and 100×.

**Figure 2 vetsci-12-00681-f002:**
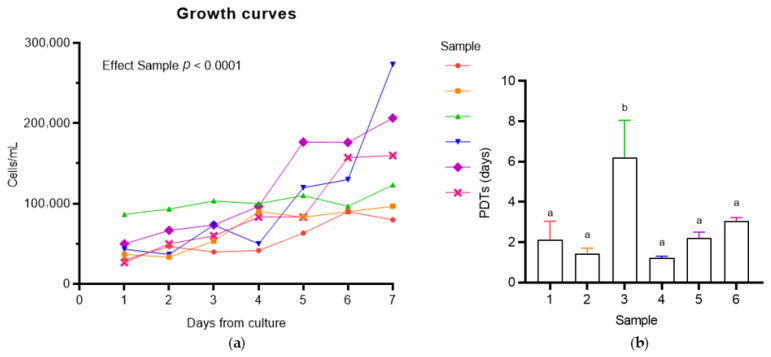
Proliferative capacity of equine colostrum-derived MSCs, evaluated through semi-logarithmic growth curves (**a**) and population doubling time (PDT) analysis (**b**). Data are presented as mean (**a**) and as mean ± SD (**b**). Statistical analysis: One-way ANOVA with Tukey’s post-hoc test (**b**). Different letters indicate significant differences (*p* < 0.05) among samples.

**Figure 3 vetsci-12-00681-f003:**
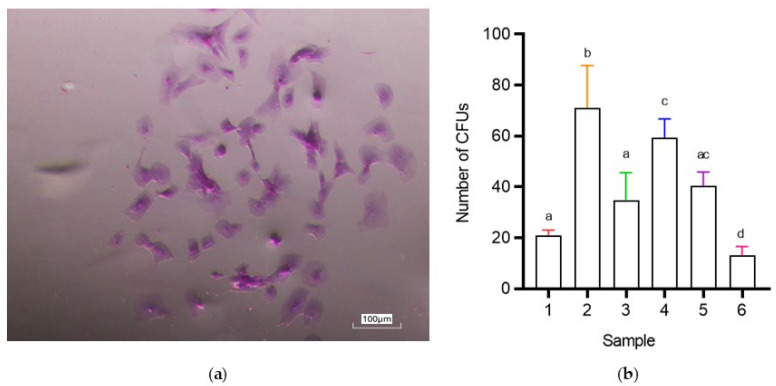
Colony-forming unit (CFU) assay of colostrum-derived MSCs. (**a**) Representative image of dispersed CFUs after Giemsa staining. Magnification: 100×. (**b**) Average number of CFUs per sample. Data are expressed as mean ± SD. Statistical analysis: One-way ANOVA with Tukey’s post-hoc test. Different letters indicate significant differences (*p* < 0.05) among samples.

**Figure 4 vetsci-12-00681-f004:**
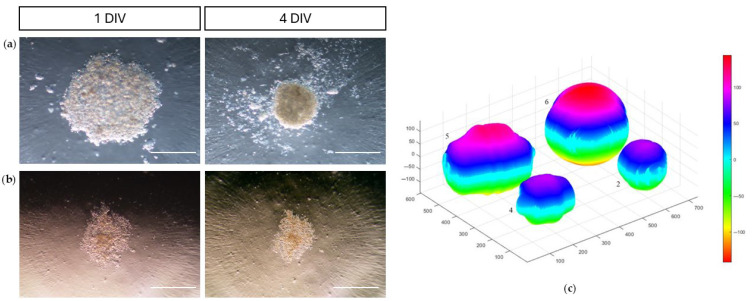
Adhesion assay of colostrum-derived MSCs. Representative bright-field images showing spheroid-forming (**a**) and non-forming (**b**) colostrum-derived MSCs after 1 to 4 days of hanging drop culture (5000 cells/25 µL drop). Magnification: 100×. DIV, day in vitro. 3D spheroid volume reconstruction was performed using *ReViSP* software (sourceforge.net/projects/revisp/, v. 2.3) (**c**), starting from the acquired 2D bright-field images. Measure unit: pixel.

**Figure 5 vetsci-12-00681-f005:**
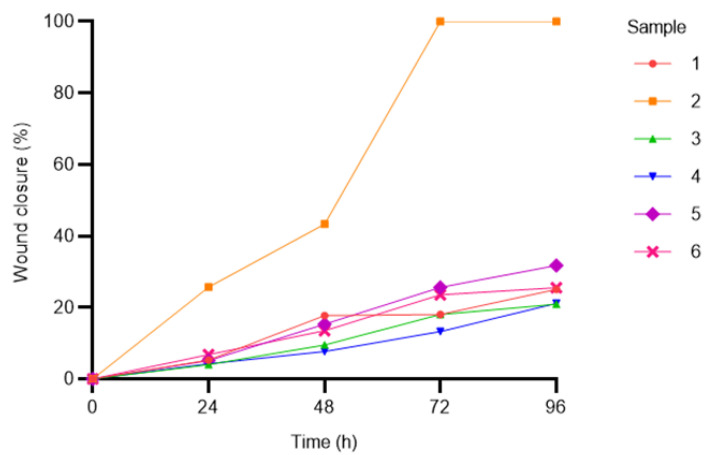
Scratch assay on colostrum-derived MSCs. Wound closure percentages were calculated using ImageJ software at 24, 48, 72, and 96 h using the following formula: [(distance at t*0* − distance at t*n* *100]/distance at t*0*. Data, presented as mean values, are plotted over time.

**Figure 6 vetsci-12-00681-f006:**
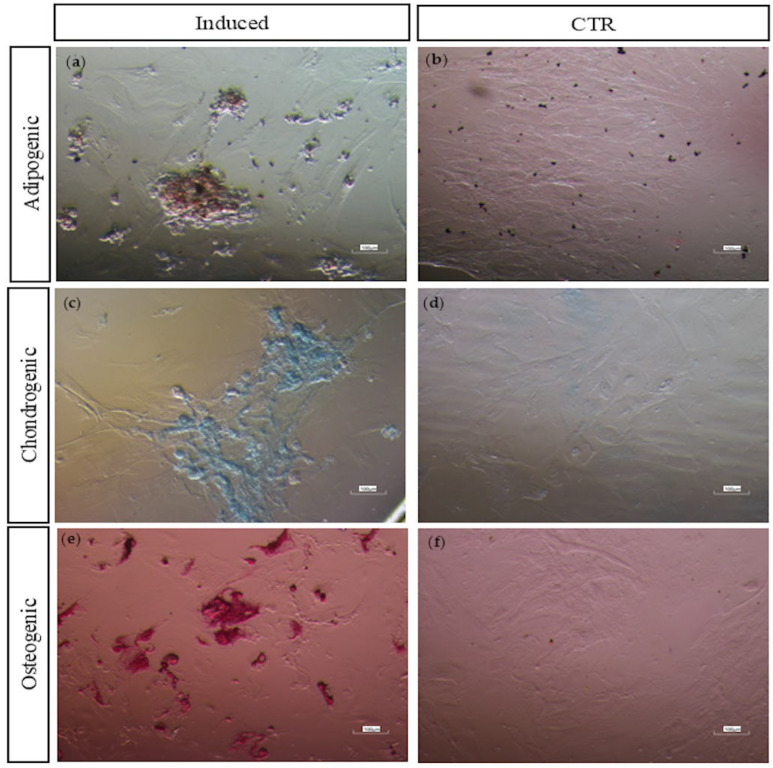
Multilineage differentiation of colostrum-derived MSCs. (**a**) Adipogenic differentiation: Representative Oil Red O staining shows intracellular lipid droplets after 3 weeks of induction. (**b**) Adipogenic control: Representative image of cells cultured in standard medium for 21 days, showing normal morphology and no staining. (**c**) Chondrogenic differentiation: Representative Alcian Blue staining highlights glycosaminoglycans in the cartilage matrix after 3 weeks of induction. (**d**) Chondrogenic control: Representative image of cells in standard medium for 21 days, showing normal morphology and no staining. (**e**) Osteogenic differentiation: Representative Alizarin Red S staining indicates extracellular calcium deposition after 3 weeks of induction. (**f**) Osteogenic control: Representative image of cells in standard medium for 21 days, showing normal morphology and no staining. Magnifications: Induction panels: 100×, controls: 40×.

**Figure 7 vetsci-12-00681-f007:**
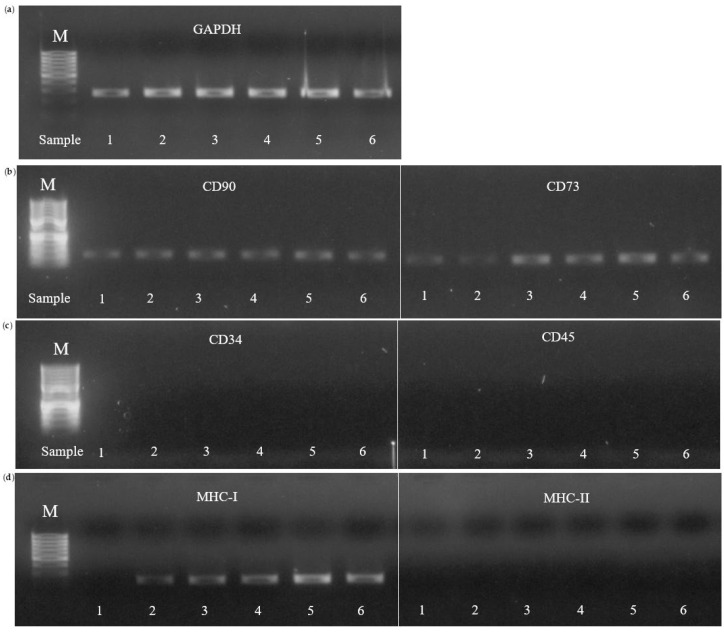
RT-PCR analysis of MSC marker expression in colostrum-derived MSCs. (**a**) Expression of the housekeeping gene GAPDH in MSC markers CD90 and CD73 (**b**), hematopoietic markers CD34 and CD45 (**c**), and major histocompatibility complex markers MHC-I and MHC-II (**d**). Lane M in each panel corresponds to the molecular weight marker (50 bp DNA ladder).

**Table 1 vetsci-12-00681-t001:** Specific induction media’s compositions.

Adipogenic	Chondrogenic	Osteogenic
DMEM	DMEM	DMEM
10% FBS	1% FBS	10% FBS
1 µM DXM ^2^ (*removed after 6 days*)	0.1 µM DXM ^2^	0.1 µM DXM ^2^
0.5 mM IBMX ^1^ *(removed after 3 days)*	50 nM AA2P ^4^	50 µM AA2P ^4^
10 µg/mL insulin	6.25 µg/mL insulin	10 mM BGP ^5^
0.1 mM indomethacin	10 ng/mL hTGF-β1 ^3^	

^1^ IBMX: isobutylmethylxanthine, ^2^ DXM: dexamethasone, ^3^ hTGF-β1: human transforming growth factor-β1, ^4^ AA2P: ascorbic acid 2-phosphate, ^5^ BGP: beta-glycerophosphate.

**Table 2 vetsci-12-00681-t002:** Primers’ sequences for PCR analysis.

Primers	References	Sequences (5′ → 3′)	bp
*MSC markers*			
CD90	[83]	F: TGCGAACTCCGCCTCTCT R: GCTTATGCCCTCGCACTTG	93
CD73	[83]	F: GGGATTGTTGGATACACTTCAAAAG R: GCTGCAACGCAGTGATTTCA	90
*Hematopoietic markers*			
CD34	[83]	F: CACTAAACCCTCTACATCATTTTCTCCTA R: GGCAGATACCTTGAGTCAATTTCA	101
CD45	[83]	F: TGATTCCCAGAAATGACCATGTA R: ACATTTTGGGCTTGTCCTGTAAC	101
*MHC markers*			
MHC-I	[84]	F: GGAGAGGAGCAGAGATACA R: CTGTCACTGTTTGCAGTCT	218
MHC-II	[85]	F: TCTACACCTGCCAAGTG R: CCACCATGCCCTTTCTG	178
*Housekeeping*			
GAPDH	[85]	F: GTCCATGCCATCACTGCCAC R: CCTGCTTCACCACCTTCTTG	262

**Table 3 vetsci-12-00681-t003:** Donor, colostrum, and colostrum-derived MSC characteristics: maternal age and number of pregnancies; colostrum volume, Brix index score, cell yield, and days to confluence; MSC population doubling time and colony-forming units.

Mare/Sample	Age (Years)	Number of Pregnancies	Volume (mL)	Brix Index (%)	Cell Yield (×10^3^ cells/mL)	Days to Confluence	Mean PDTs ^1^ (Days)	Number of CFUs ^2^
1	11	5	25	25	260	10	2.1 ± 0.9	21 ± 2
2	20	14	23	29.5	30	10	1.4 ± 0.3	71 ± 16.7
3	17	2	17	27	300	11	6.2 ± 1.8	34.7 ± 11
4	13	4	20	26	500	8	1.2 ± 0.1	59.3 ± 7.4
5	8	1	21	35	480	11	2.2 ± 0.3	40.3 ± 5.5
6	14	3	11	28	400	11	3.1 ± 0.2	13 ± 3.6
Mean	14.8 ± 4.26	5.17 ± 4.71	19.5 ± 4.97	28.2 ± 3.96	328.3 ± 174.4	10.17 ± 1.17	2.7 ± 0.6	39.9 ± 7.7

^1^ PDTs: population doubling times, ^2^ CFU: colony-forming unit.

**Table 4 vetsci-12-00681-t004:** Results of PCR analysis of MSCs derived from six colostrum samples at P3.

Sample	MSC Markers		Hematopoietic Markers		MHC Markers	
	CD 90	CD73	CD34	CD 45	MHC-I	MHC-II
1	+	+	-	-		-
2	+	+	-	-	+/−	-
3	+	+	-	-	+/−	-
4	+	+	-	-	+/−	-
5	+	+	-	-	+/−	-
6	+	+	-	-	+/−	-

## Data Availability

The data presented in this study are contained within the article.

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
