# Peer review of "Equine Colostrum-Derived Mesenchymal Stromal Cells: A Potential Resource for Veterinary Regenerative Medicine"

_vetsci, 2025, doi:10.3390/vetsci12070681_

Round 1
Reviewer 1 Report
Comments and Suggestions for Authors
The technique is well described. I would add a list with the abbreviations or be sure that if the author uses an abbreviation for the first time in the manuscript that the abbreviation is explained.
Author Response
We would like to thank the reviewer for the time dedicated to reviewing our manuscript and for the constructive comments provided. Please find below our point-by-point responses.
Comment 1:
The technique is well described. I would add a list with the abbreviations or be sure that if the author uses an abbreviation for the first time in the manuscript that the abbreviation is explained.
Response 1: We would like to thank the reviewer for the time dedicated to reviewing our manuscript and for the constructive comments provided. As suggested by the reviewer, we added a list with the abbreviations used in the text.

Reviewer 2 Report
Comments and Suggestions for Authors
Dear authors,
I have read your manuscript with joy and interest. It is well written, easy to understand. I don't think that major improvements are necessary or could be advised. I only have got a few minor questions/suggestions.
What is the difference between mesenchymal stem cells and mesenchymal stromal cells? (cfr. line 455 where both seem to be used as synonyms)
Line 31: PDT? (is only described in full in line 180)
Line 327: “ad” should be “as”
Lines 330-332: “Colony-forming Unit” vs. Colony Forming Unit”
Line 346: Are you sure that the magnification is 10x? Not 10x objective lens and 10x ocular lens which make 100x? No need to mention that the scale bar is 100 µm because it is on the picture. (idem Fig. 4 and 6)
Author Response
We would like to thank the reviewer for the time dedicated to reviewing our manuscript and for the constructive comments provided. Please find below our point-by-point responses. We note that the specific textual changes made to the manuscript are not included in this response letter; instead, they are clearly marked in the revised manuscript file and we have indicated here as "see manuscript file" in the corresponding response.
Comment 1
Dear authors,
I have read your manuscript with joy and interest. It is well written, easy to understand. I don't think that major improvements are necessary or could be advised. I only have got a few minor questions/suggestions.
Response 1: The authors sincerely thank the reviewer for this comment.
Comment 2:
What is the difference between mesenchymal stem cells and mesenchymal stromal cells? (cfr. line 455 where both seem to be used as synonyms)
Response 2:
The terms "mesenchymal" and "stromal" are often used interchangeably and are considered synonyms in many scientific contexts, especially when referring to multipotent cells derived from connective tissues. However, in recent years, "stromal" has become the more appropriate and widely accepted term. This shift reflects a more precise understanding of the cells' origin and function. While "mesenchymal" suggests an embryonic lineage, "stromal" better captures the supportive, structural role these cells play within adult tissues. Therefore, both terms are technically correct. In any case, given the reviewer’s comment, and to avoid causing confusion for the reader, the authors have decided to refer to the cells under study as Mesenchymal Stem/Stromal Cells throughout the manuscript.
Comment 3:
Line 31: PDT? (is only described in full in line 180)
Response 3: See manuscript file
Comment 4:
Line 327: “ad” should be “as”
Response 4: See manuscript file
Comment 5:
Lines 330-332: “Colony-forming Unit” vs. Colony Forming Unit”
Response 5: See manuscript file
Comment 6:
Line 346: Are you sure that the magnification is 10x? Not 10x objective lens and 10x ocular lens which make 100x? No need to mention that the scale bar is 100 µm because it is on the picture. (idem Fig. 4 and 6)
Response 6: See manuscript file

Reviewer 3 Report
Comments and Suggestions for Authors
Simple summary and abstract offer an appropriate overview of the work.
All the while being an abstract, you should use the extended description of any new name as in Line 31 PDT hasn't been mentioned earlier and reader will have to go lower in the text to know what it describes, same for acronym CFU.
Keywords: the title is already containing all the keywords, possibly add some different.
The introduction gives an excellent summary of the current knowledge about C-MSC and a good overview about MSC definition. You state the aim of your research and you could possiblly articulate an hypothesis to introduce the aspect of your research proving you can isolate MSC from equine colostrum.
In your material and method, the breed (B), age (A) and number of pregnancies (P#), physiologic status especially with regards to the genital system, and grade of risk of the pregnancy of the mares used for your project should be included and possibly included in the analyse of your results. Other research have demonstrated the huge disparity of the concentration of MSC's between horses and it would be useful to know if any of the parameters (B, A, P#,...) may or may not have an influence on your measurements in particular cell yield, PTD and CFU.
Line 139, with respect to word count, can you describe in more details how you perform an aseptic collection of colostrum if by stripping, or did you perform a direct aspiration with needle?
Lines 153 + 159, I presume the concentration of streptomycin is 100ug/mL not 100 g/mL
Line 196 you mention RT without precision it is room temperature but does the opposite at line 233, consistency I recommend inverting these
Line 294 vs table 3: there is a huge discrepancy between the text and the numbers on the table, can you explain this gap.
Line 351, not sure you have to repeat the method used to assess cell-cell adhesion here
Line 368 : same observation that you repeat the method used and its rationale while you should have addressed this in M&M or in the discussion
Line 378: figure 5 doesn't illustrate this statement, are you sure it is sample 3 and not sample 5?
Line 381, you could put the actual p value
Line 393 same comment as 351 and 368
Line 430 idem: you can skip the first paragraph and my win some word count if necessary
Line 472, as you have not compared with later lactation milk in the present project, I think that using optimal here isn't appropriate
Line 475, review grammar of this sentence, you've got a verb too much or a linking word too few
You very appropriately underscore the variability of your findings leaving aside any consideration of the donor variable, your paper would gain in strength if the different characteristics of your donor mares were included in your statistical analysis - correlation might not be causation as your sample is too small to establish this aspect but you would increase the validity of your findings with this inclusion.
I understand your enthusiasm but nothing in your paper addresses or support the last sentence of your conclusion. Promising should at the least be replaced by potential or possible translational implications ...
Author Response
We would like to thank the reviewer for the time dedicated to reviewing our manuscript and for the constructive comments provided. Please find below our point-by-point responses. We note that the specific textual changes made to the manuscript are not included in this response letter; instead, they are clearly marked in the revised manuscript file and we have indicated here as "see manuscript file" in the corresponding response.
Comment 1:
Simple summary and abstract offer an appropriate overview of the work.
All the while being an abstract, you should use the extended description of any new name as in Line 31 PDT hasn't been mentioned earlier and reader will have to go lower in the text to know what it describes, same for acronym CFU.
Response 2: See manuscript file.
Comment 2:
Keywords: the title is already containing all the keywords, possibly add some different.
Response 2: See the manuscript file
Comment 3:
The introduction gives an excellent summary of the current knowledge about C-MSC and a good overview about MSC definition. You state the aim of your research and you could possibly articulate an hypothesis to introduce the aspect of your research proving you can isolate MSC from equine colostrum.
Response 3: See the manuscript file
Comment 4: In your material and method, the breed (B), age (A) and number of pregnancies (P#), physiologic status especially with regards to the genital system, and grade of risk of the pregnancy of the mares used for your project should be included and possibly included in the analyse of your results. Other research have demonstrated the huge disparity of the concentration of MSC's between horses and it would be useful to know if any of the parameters (B, A, P#,...) may or may not have an influence on your measurements in particular cell yield, PTD and CFU.
Response 4: The authors thank the reviewer for the comment. As reported in the manuscript (Line 137), all the mares used in the study were of the same breed and healthy. Other data have been reported in Table 3 and the correlations between age, parity, cell yeld and population doubling time have been evaluated as suggested by the reviewer and reported in the results and discussion section as highlighted in the manuscript file.
Comment 5:
Line 139, with respect to word count, can you describe in more details how you perform an aseptic collection of colostrum if by stripping, or did you perform a direct aspiration with needle?
Response 5: See the manuscript file
Comment 6:
Lines 153 + 159, I presume the concentration of streptomycin is 100ug/mL not 100 g/mL
Response 6: See the manuscript file
Comment 7:
Line 196 you mention RT without precision it is room temperature but does the opposite at line 233, consistency I recommend inverting these
Response 7: See the manuscript file
Comment 8:
Line 294 vs table 3: there is a huge discrepancy between the text and the numbers on the table, can you explain this gap.
Response 8: See the manuscript file
Comment 9:
Line 351, not sure you have to repeat the method used to assess cell-cell adhesion here
Response 8: See the manuscript file
Comment 10:
Line 368: same observation that you repeat the method used and its rationale while you should have addressed this in M&M or in the discussion
Response 10: See the manuscript file
Comment 11:
Line 378: figure 5 doesn't illustrate this statement, are you sure it is sample 3 and not sample 5?
Response 11: See the manuscript file
Comment 12:
Line 381, you could put the actual p value
Response 12: See the manuscript file
Comment 13:
Line 393 same comment as 351 and 368
Response 13: See the manuscript file
Comment 14:
Line 430 idem: you can skip the first paragraph and my win some word count if necessary
Response 14: See the manuscript file
Comment 15:
Line 472, as you have not compared with later lactation milk in the present project, I think that using optimal here isn't appropriate
Response 15: See the manuscript file
Comment 16:
Line 475, review grammar of this sentence, you've got a verb too much or a linking word too few
Response 16: See the manuscript file
Comment 18:
You very appropriately underscore the variability of your findings leaving aside any consideration of the donor variable, your paper would gain in strength if the different characteristics of your donor mares were included in your statistical analysis - correlation might not be causation as your sample is too small to establish this aspect but you would increase the validity of your findings with this inclusion.
Response 18: Please see the response to comment 4 and the manuscript file.
Comment 19:
I understand your enthusiasm but nothing in your paper addresses or support the last sentence of your conclusion. Promising should at the least be replaced by potential or possible translational implications ...
Response 19: See the manuscript file

Reviewer 4 Report
Comments and Suggestions for Authors
The submitted manuscript “Equine Colostrum-derived Mesenchymal Stromal Cells: A Potential Resource for Veterinary Regenerative Medicine” focus on an important source of MSCs to the equine regenerative medicine. The demand of different techniques in equine regenerative medicine has been increasing and the study of novel, non-invasive methods can be very interesting.
In general, the idea of study and conceptualization are interesting. The low number of horses evaluated imposes caution in the results (even with statistical adaptations to that number).
The Title is adequate and reflects the subject of the manuscript.
The Abstract is satisfactory and well written. It provides information regarding the content of the manuscript (objectives, materials and methods, results).
PDT and CFU should be replaced by “Population Doubling Time (PDT)” and “colony-forming unit (CFU)”, respectively.
The keywords are adequate to the content of the study.
The introduction highlights the importance of the theme and provides correct known information regarding the subjects studied.
The methodology is well described and suitable for the objectives.
The results are correctly expressed.
In the legend of table 3, the meaning of PDTs and CFU should be included.
The Discussion is satisfactory.
The authors discuss the characteristics of C-MSCs and give information about the potential use in regenerative medicine.
The conclusions are sound considering the result of the study.
Author Response
We would like to thank the Reviewer for the time dedicated to reviewing our manuscript and for the constructive comments provided. Please find below our point-by-point responses. We note that the specific textual changes made to the manuscript are not included in this response letter; instead, they are clearly marked in the revised manuscript file. For each change, we have indicated "see manuscript file" in the corresponding response.
Comment 1:
The submitted manuscript “Equine Colostrum-derived Mesenchymal Stromal Cells: A Potential Resource for Veterinary Regenerative Medicine” focus on an important source of MSCs to the equine regenerative medicine. The demand of different techniques in equine regenerative medicine has been increasing and the study of novel, non-invasive methods can be very interesting.
Response 1: The authors thank the reviewer for the comment.
Comment 2:
In general, the idea of study and conceptualization are interesting. The low number of horses evaluated imposes caution in the results (even with statistical adaptations to that number).
Response 2: The authors thank the reviewer for the comment
Comment 3:
The Title is adequate and reflects the subject of the manuscript.
Response 3: The authors thank the reviewer for the comment
Comment 4:
The Abstract is satisfactory and well written. It provides information regarding the content of the manuscript (objectives, materials and methods, results).
PDT and CFU should be replaced by “Population Doubling Time (PDT)” and “colony-forming unit (CFU)”, respectively.
Response 4: See the manuscript file
Comment 5:
The keywords are adequate to the content of the study.
Response 5: the keywords are modified and implemented due to the comments of Reviewer 2.
Comment 6:
The introduction highlights the importance of the theme and provides correct known information regarding the subjects studied.
Response 6: The authors thank the reviewer for the comment
Comment 7:
The methodology is well described and suitable for the objectives.
Response 7: The authors thank the reviewer for the comment
Comment 8:
The results are correctly expressed.
Response 8: The authors thank the reviewer for the comment
Comment 9:
In the legend of table 3, the meaning of PDTs and CFU should be included.
Response 9: see the manuscript file
Comment 10:
The Discussion is satisfactory.
The authors discuss the characteristics of C-MSCs and give information about the potential use in regenerative medicine.
Response 10: The authors thank the reviewer for the comment
Comment 11:
The conclusions are sound considering the result of the study.
Response 11: The authors thank the reviewer for the comment
